# Native mass spectrometry combined with enzymatic dissection unravels glycoform heterogeneity of biopharmaceuticals

Therese Wohlschlager [1,2], Kai Scheffler[2,3], Ines C. Forstenlehner[1,2,4], Wolfgang Skala[1,2], Stefan Senn[1,2], Eugen Damoc[5], Johann Holzmann[2,4] & Christian G. Huber [1,2]

Robust manufacturing processes resulting in consistent glycosylation are critical for the efficacy and safety of biopharmaceuticals. Information on glycosylation can be obtained by conventional bottom–up methods but is often limited to the glycan or glycopeptide level. Here, we apply high-resolution native mass spectrometry (MS) for the characterization of the therapeutic fusion protein Etanercept to unravel glycoform heterogeneity in conditions of hitherto unmatched mass spectral complexity. Higher spatial resolution at lower charge states, an inherent characteristic of native MS, represents a key component for the successful revelation of glycan heterogeneity. Combined with enzymatic dissection using a set of proteases and glycosidases, assignment of specific glycoforms is achieved by transferring information from subunit to whole protein level. The application of native mass spectrometric analysis of intact Etanercept as a fingerprinting tool for the assessment of batch-to-batch variability is exemplified and may be extended to demonstrate comparability after changes in the biologic manufacturing process.

[1] Department of Biosciences, Bioanalytical Research Labs, University of Salzburg, Hellbrunner Strasse 34, 5020 Salzburg, Austria. [2] Christian Doppler Laboratory for Innovative Tools for Biosimilar Characterization, University of Salzburg, Hellbrunner Strasse 34, 5020 Salzburg, Austria. [3] Thermo Fisher Scientific GmbH, Dornierstraße 4, 82110 Germering, Germany. [4] Technical Development Biosimilars, Global Drug Development, Novartis, Sandoz GmbH, Biochemiestrasse 10, 6250 Kundl, Austria. [5] Thermo Fisher Scientific GmbH, Hanna-Kunath-Strasse 11, 28199 Bremen, Germany. Correspondence and requests for materials should be addressed to C.G.H. (email: c.huber@sbg.ac.at)

The approval of biopharmaceuticals requires in-depth characterization at the molecular level involving confirmation of the amino-acid sequence as well as comprehensive assessment of post-translational modifications (PTMs)[1]. Consistent glycosylation constitutes an important quality attribute as it may impact the efficacy and safety of the therapeutic product. Current analytical methods comprise the characterization of released glycans or of glycopeptides, providing detailed information on the average molecular composition[2,3]. Alternatively, the existence and relative abundance of specific proteoforms can be revealed by the determination of intact protein masses[4], as has been accomplished for several therapeutic monoclonal antibodies (mAbs)[5–9]. Conventional mass determination under denaturing conditions is restricted to samples of limited complexity due to the overlap of broad signal clusters observed for the respective charge states. In contrast, under the conditions applied in native mass spectrometry (MS), protein folding is preserved resulting in lower charge states situated at higher $m/z$ in the corresponding mass spectrum. The inherently higher spatial resolution at high $m/z$ allows separation and thus deconvolution of these broad signal clusters. This technique is well established for the characterization of non-covalent protein complexes, e.g., large protein assemblies[10–13] or antibody–drug conjugates[14,15]. Native MS has also been applied for qualitative and semi-quantitative analysis of composite mixtures of mAbs[16,17] as well as for characterization of micro-heterogeneity in therapeutic proteins arising from, for example, $N$-glycosylation variants[18–22]. In this context, the glycan heterogeneity of human erythropoietin (26–30 kDa) and human plasma properdin (54 kDa) was successfully revealed upon integration of intact protein mass determination by native MS, middle–down analysis of proteolytic glycopeptides, and enzymatic deglycosylation, which facilitated the assignment of PTM compositions to the detected intact protein masses[22].

Etanercept, the active pharmaceutical ingredient of Enbrel®, is a highly glycosylated therapeutic Fc-fusion protein. This biopharmaceutical acts as an inhibitor for tumor necrosis factor (TNF), an important mediator protein of inflammatory cell responses, and has been licensed for the treatment of autoimmune disorders such as rheumatoid and psoriatic arthritis[23].

Etanercept consists of a TNF-α receptor (TNFR) domain fused to the Fc portion of human IgG1 and forms dimers stabilized by three intermolecular disulfide bonds resulting in a theoretical protein mass of 102.4 kDa (Fig. 1a; for amino-acid sequence see Supplementary Fig. 1)[24]. The resultant protein comprises multiple glycosylation sites: four $N$- and 26 $O$-glycosylation sites in the dimeric TNFR domain, as well as two $N$-glycan sites in the Fc domain, as previously characterized via released $N$- and $O$-glycan analysis by hydrophilic interaction liquid chromatography with fluorescence detection[24]. In the same study, $O$-glycopeptides were analyzed by HPLC-MS with collision-induced- and electron transfer dissociation. The $O$-glycans were found to be predominantly of the core 1 subtype (Galβ1-3GalNAc-) substituted by up to two sialic acid residues ($N$-acetylneuraminic acid, Neu5Ac)[24].

Here, we report on the applicability of native MS for resolution of the highly complex glycosylation pattern of the 130 kDa therapeutic Fc-fusion protein Etanercept. Extending previous work on smaller proteins[22] we take advantage of the superior spatial resolution for protein isoforms at lower charge states detectable under native conditions in order to uncover glycoforms and explore the limits of mass spectral resolution and mass accuracy with glycoproteins larger than 100 kDa. To reduce spectral complexity and hence facilitate the annotation of glycoforms and other proteoforms, we employ enzymatic dissection using specific proteases and glycosidases. Our approach serves to provide comprehensive information on both the $N$- and $O$-glycosylation patterns at different levels of molecular complexity. We also demonstrate that molecular information gained at lower structural levels (i.e., glycopeptides, protein subunits) can successfully be integrated to facilitate glycoform annotation at higher structural levels (i.e., whole protein upon partial deglycosylation) through the application of advanced computational tools. Thus, the obtained molecular mass information at the whole protein level allows us to propose of the most likely combination of glycan structures. Finally, native MS of Etanercept may serve as a rapid fingerprinting tool for the assessment of batch-to-batch variability at the intact protein level.

## Results

**Native mass spectrometry of Etanercept at the intact level.** First, we assessed the feasibility of intact mass measurements of

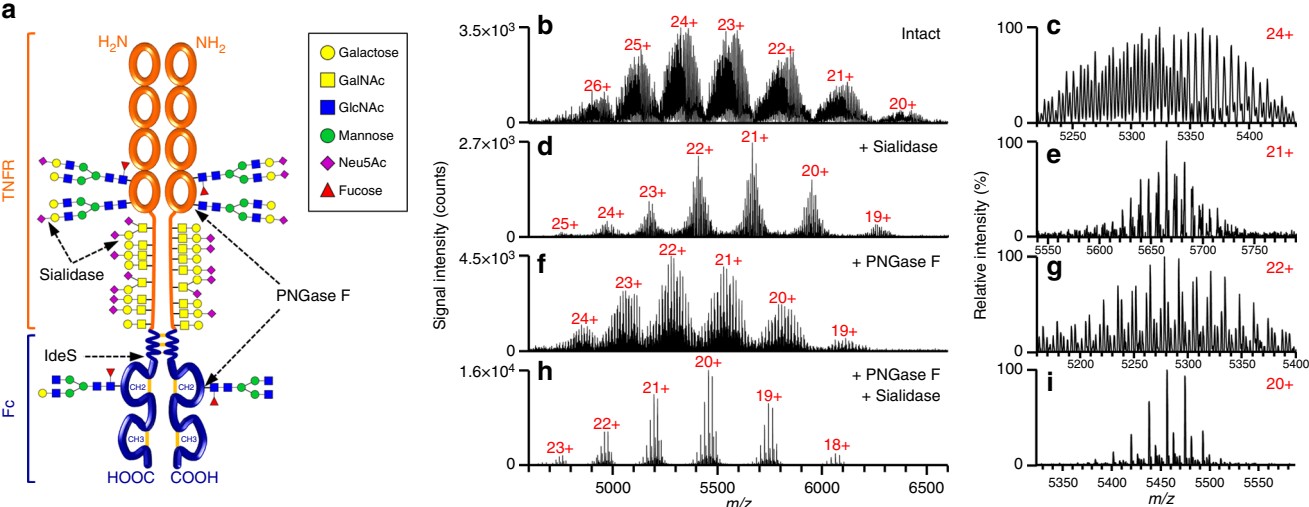

**Fig. 1** Molecular structure and native mass spectrometry of Etanercept. **a** Schematic illustration of dimeric Etanercept consisting of a TNFR and an Fc domain. Disulfide bonds in the Fc region are indicated as yellow lines; disulfide bridges in the TNFR domain are not shown. Monosaccharide symbols are listed. Exemplary cleavage sites of IdeS, PNGase F and sialidase are indicated. Native mass spectra of **b** intact Etanercept ($R_{set} = 17,500$ at $m/z$ 200), **d** Etanercept digested with sialidase or **f** PNGase F ($R_{set} = 35,000$ at $m/z$ 200), and **h** a combination of PNGase F/sialidase, respectively ($R_{set} = 70,000$ at $m/z$ 200). Charge states are indicated. Zooms into the most abundant charge states are shown in **c**, **e**, **g**, and **i**

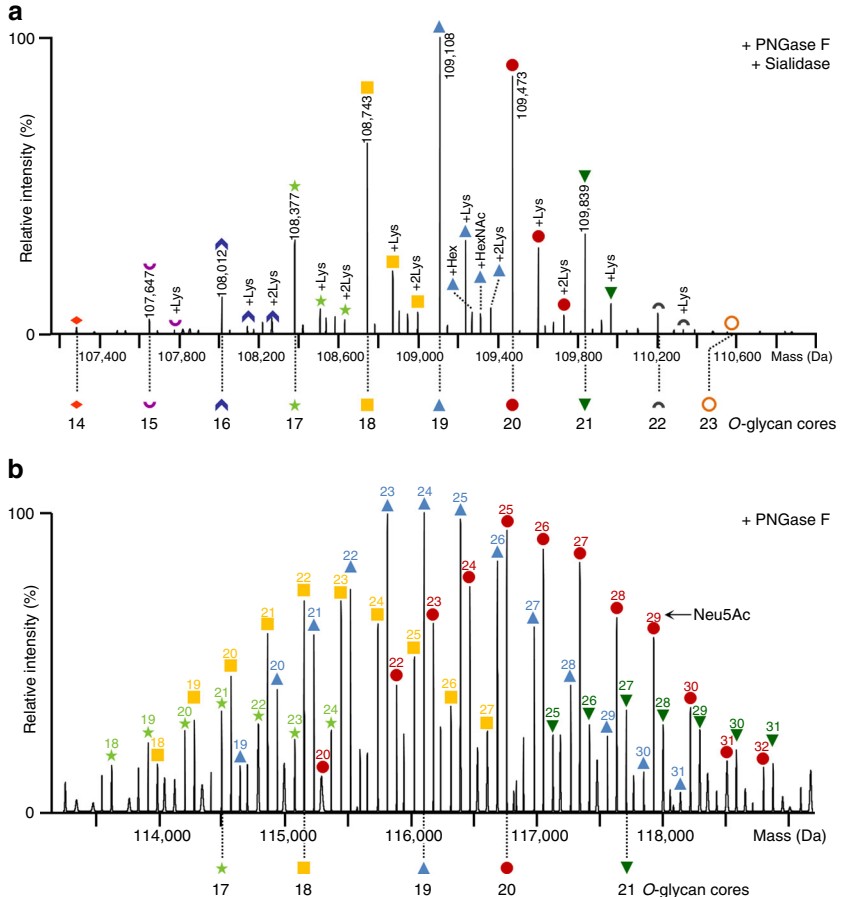

**Fig. 2** Annotation of Etanercept O-glycoforms. **a** Deconvoluted spectrum of Etanercept after digestion with PNGase F/sialidase (raw spectrum shown in Fig. 1h). O-glycoforms, i.e., the number of core 1 units (Hex-HexNAc), as well as lysine variants are annotated. Each symbol indicates a certain number of O-glycan cores. **b** Deconvoluted spectrum of Etanercept after digestion with PNGase F (raw spectrum shown in Fig. 1f). The number of O-glycan cores is indicated by a specific symbol in accordance with Fig. 2a. Multiple signals annotated with the same symbol represent sialic acid (Neu5Ac) variants of each O-glycoform. The number of Neu5Ac residues is indicated above each annotated peak. Peak lists with all possible glycoform assignments are available in Supplementary Data 1

Etanercept on a Thermo Scientific™ Exactive™ Plus EMR mass spectrometer. Distinctive features of this instrument include (i) an extended mass range of m/z 350-20,000 and (ii) the option of trapping ions in the higher energy collisional dissociation (HCD) cell at adjustable gas pressure. For the analysis, Enbrel® drug product was extensively buffer exchanged to ammonium acetate and directly infused by static nano-ESI. We observed a charge state distribution from m/z 4800 to 6500 corresponding to charges of 26+ to 20+ on the intact molecule (Fig. 1b). Each charge state consisted of numerous signals, potentially arising from different proteoforms including glycan variants, as illustrated in the zoom of charge state 24+ (Fig. 1c). The superior spatial resolution of native MS was a prerequisite to this analysis, as the charge states typically obtained for such complex proteins under denaturing conditions (50+ to 35+) would overlap (simulation shown in Supplementary Fig. 2).

Deconvolution of the raw spectrum of intact Etanercept using the ReSpect™ algorithm (embedded in the Thermo Scientific™ BioPharma Finder™ software) resulted in masses ranging from 125 to 131 kDa for the intact protein variants (see Supplementary Fig. 3). The multitude of signals and their relative abundances observed in the raw data obtained with an instrument resolution setting (abbreviated as $R_{set}$ in the following) of 17,500 at m/z 200 (see Fig. 1c and Supplementary Fig. 3a) is clearly reflected in at least 70 distinguishable protein

signals in the deconvoluted spectrum (Supplementary Fig. 3b). Taking into account a theoretical average mass of 102,160 Da for the total Etanercept amino-acid sequence (dimer without C-terminal lysine residues, 29 disulfide bonds[25]), the experimentally determined masses evidence residual masses of ~ 23,000 to 29,000 Da, corresponding to the glycan portion of the protein.

By virtue of the vast number of glycan combinations possible from structures reported for Etanercept to date[24], the intact Etanercept spectra were too complex to assign specific glycoforms to the residual masses. Subsequently, a systematic strategy applying glycosidases and/or proteases was employed to successively decrease the complexity and molecular size of the sample. Ultimately, this strategy aimed at assembling all modifications present in Etanercept, as determined at different structural levels. Thus, we performed (i) de-sialylation with sialidase, (ii) de-N-glycosylation with PNGase F, (iii) a combination of both, (iv) double digestion with sialidase and O-glycosidase, (v) decomposition of Etanercept into TNFR and Fc domain by IdeS digestion, and (vi) digestion into glycopeptides using trypsin or AspN, followed by mass spectrometric analysis of the obtained products.

**Determination of O-glycoforms upon removal of N-glycans.** Taking advantage of the substrate specificity of different

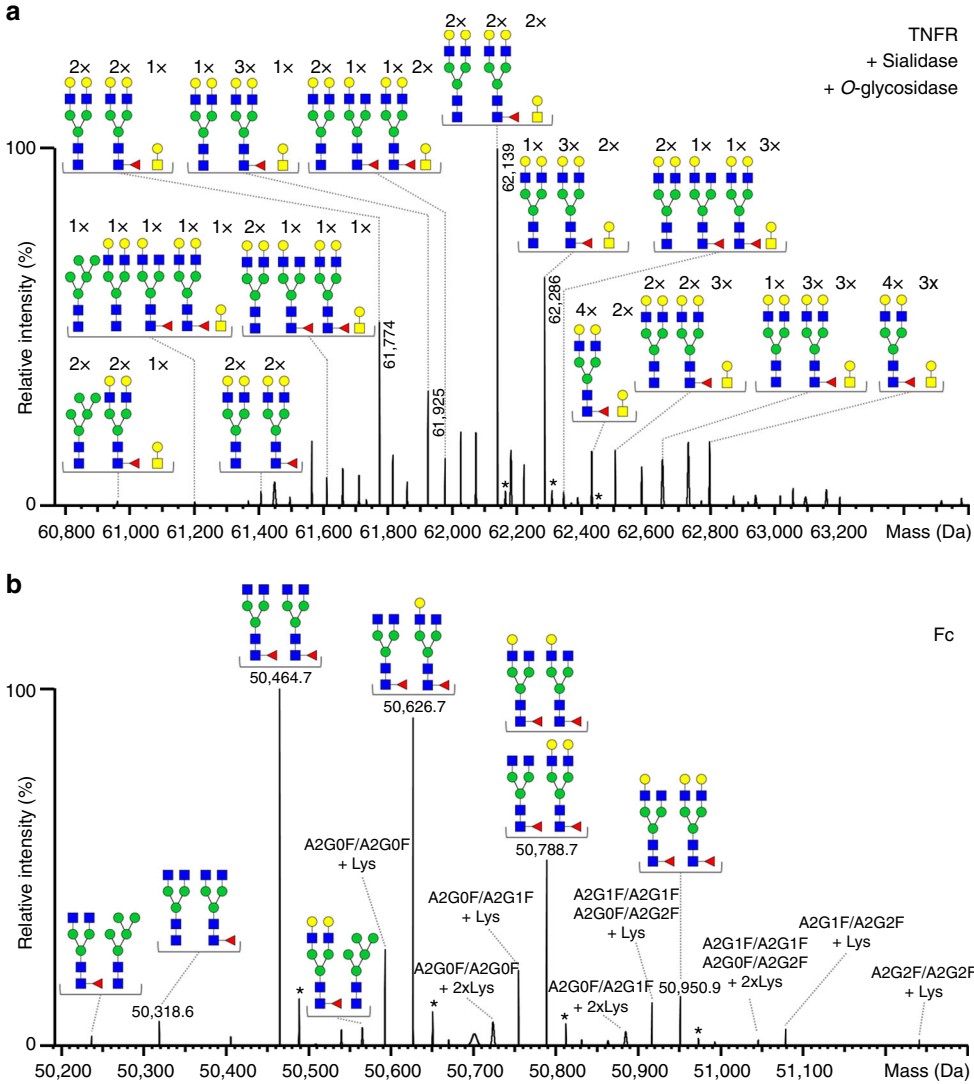

**Fig. 3** *N*-glycosylation of Etanercept TNFR and Fc domains. **a** Deconvoluted spectrum of dimeric TNFR digested with sialidase and *O*-glycosidase acquired under native conditions (raw spectrum shown in Supplementary Fig. 6c, d). The most probable glycan structures lacking sialic acids are annotated. **b** Deconvoluted spectrum of Fc dimer upon native MS (raw spectrum shown in Supplementary Fig. 7). The most probable *N*-glycoforms and C-terminal lysine variants are annotated. Asterisks indicate Na$^+$ adducts. Peak lists with all possible glycoform assignments are available in Supplementary Data 1

glycosidases, we removed the *N*-glycans using PNGase F, sialic acids using sialidase, or both in a combined enzymatic digest (see Fig. 1a; resulting schematic protein structures are shown in Supplementary Fig. 4). Mass spectrometric analysis expectedly showed a shift of signals toward lower *m/z*, as well as reduced complexity, after glycosidase digestion (Fig. 1d–i). Removal of both the *N*-glycans and sialic acid residues resulted in glyco-forms which differed only by the number of *O*-glycan cores (Fig. 1h, i; Supplementary Fig. 4). Consequently, the most abundant peaks in the deconvoluted spectrum of the double glycosidase digest differed by a mass of 365.1 Da, attributable to a hexose (Hex) and an *N*-acetylhexosamine (HexNAc) residue, characteristic of a core 1 *O*-glycan structure (Fig. 2a). Considering multiples of core 1 *O*-glycan masses, we were able to assign the specific *O*-glycosylation variants substituted with 14–23 core 1 units as shown in Fig. 2a, using a custom software tool (MoFi)[26]. For each *O*-glycoform, variants with one or two lysine residues as well as an additional hexose or HexNAc residue were detected. Presence of lysine variants is common in Fc domain containing proteins and results from incomplete clipping of C-terminal lysine residues from the heavy chain by

carboxypeptidase B in mammalian protein expression systems[27]. Owing to the lower complexity, and consequent lower number of signals per charge state, mass spectrometric analysis of the above described double digest could also be performed under denaturing conditions (Supplementary Fig. 5). Indeed, the masses and relative abundances determined under native and denaturing conditions were highly consistent in both mass spectrometric approaches (Supplementary Fig. 5e).

Compared with the PNGase F/sialidase double digest, the spectrum of PNGase F treated Etanercept displays a higher complexity, arising from the sialic acid variants of *O*-glycoforms (Fig. 1f, g; Supplementary Fig. 4), visible as a series of signals differing in molecular mass by 291.3 Da for a given number of attached *O*-glycan cores (Fig. 2b). In a purely mass-based approach, i.e., if relative abundances from the spectrum shown in Fig. 2a were not taken into account, two possible glycoforms could be assigned for each peak in the deconvoluted spectrum within the accepted mass tolerance of ±5 Da. Considering the relative abundances of *O*-glycoforms as observed in the double glycosidase digest (Fig. 2a), sialylated variants of the corresponding *O*-glycoforms can be annotated (Fig. 2b). Thus, the most

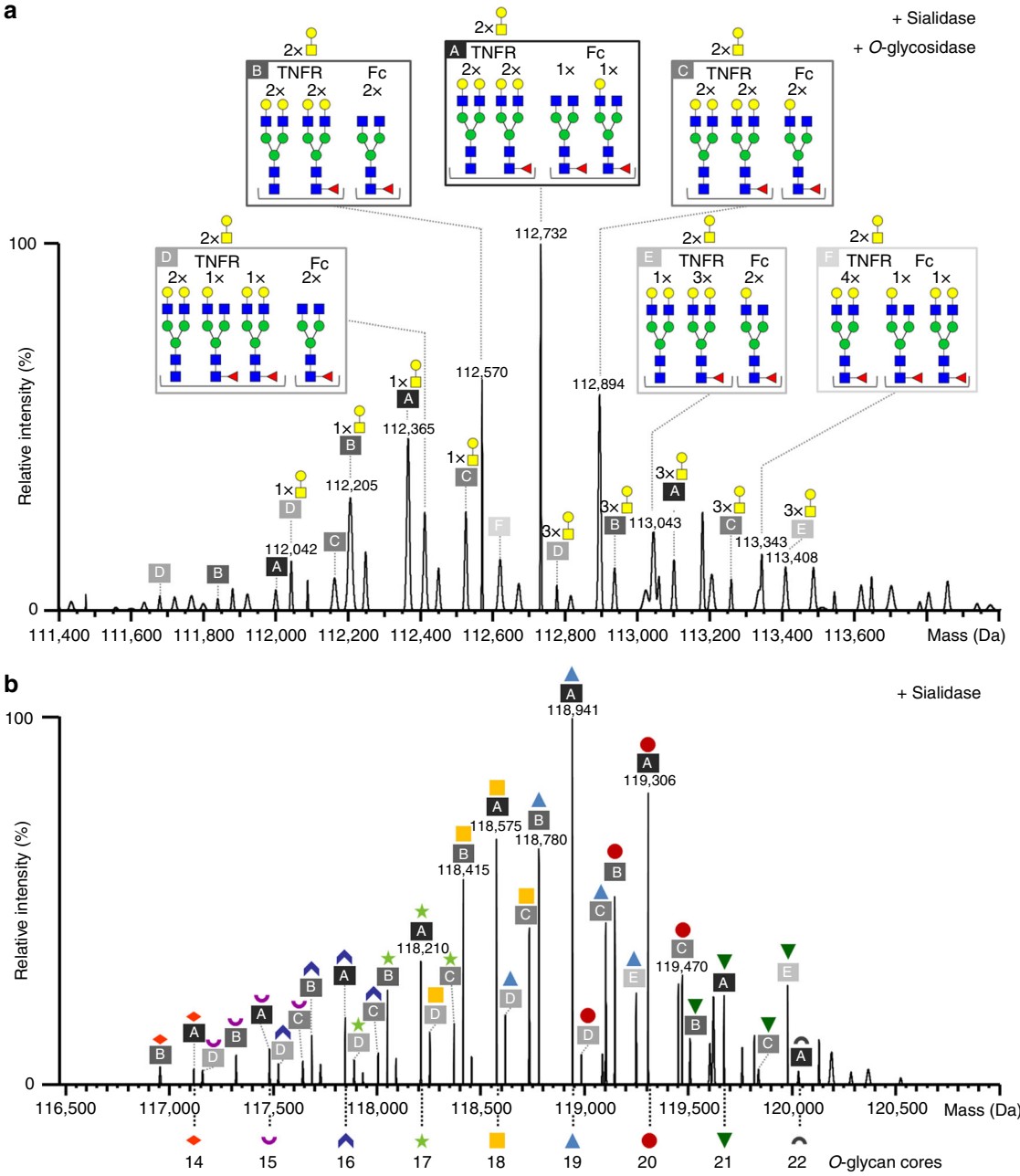

**Fig. 4** *N*- and *O*-glycosylation of Etanercept lacking sialic acids. **a** Deconvoluted spectrum of Etanercept treated with sialidase and *O*-glycosidase acquired under native conditions (raw spectrum is shown in Supplementary Fig. 8b). The most probable glycan structures lacking sialic acids are annotated. The six most abundant *N*-glycoforms are boxed and marked as A to F, respectively. **b** Deconvoluted spectrum of sialidase-treated Etanercept upon native MS (raw spectrum is shown in Fig. 1d). The most probable glycoforms are annotated. *N*-glycan structures are referred to as A to F as specified in Fig. 4a; *O*-glycoforms are labeled according to Fig. 2a. Peak lists with all possible glycoform assignments are available in Supplementary Data 1

abundant sialylation variants arise from *O*-glycoforms substituted with 18–21 core 1 *O*-glycan units (yellow squares, blue triangles, red circles, and dark green triangles in Fig. 2b). However, this abundance-based approach remains insufficient for unambiguous annotation of a series of sialylation variants observed between 113,500 and 115,500 Da; the respective series may be explained by both 17 and 21 *O*-core variants substituted with different numbers of sialic acids. As both the 17 and 21 *O*-glycoforms are of similar abundance in the double glycosidase digest (light green stars: 31.7% and dark green triangles: 33.7%; Fig. 2a), discrimination between these two alternatives is not possible. Given that the most abundant sialylated variants of

each *O*-glycoform in the PNGase F digest consistently display between 1.2 and 1.3 as a characteristic ratio of Neu5Ac residues to *O*-glycan cores, a unique annotation for this series can be attributed to 17 *O*-glycan cores (light green stars in Fig. 2b). Furthermore, series of smaller signals shifted by ~128 Da present in the deconvoluted spectrum of PNGase F digested Etanercept most probably signify the C-terminal lysine variants. The 21 Neu5Ac variant of the glycoform series substituted with 20 core 1 *O*-glycans (red circles, Fig. 2b) is not annotated owing to a mass deviation outside of the accepted mass tolerance window (±5 Da), most likely owing to interference with another proteoform.

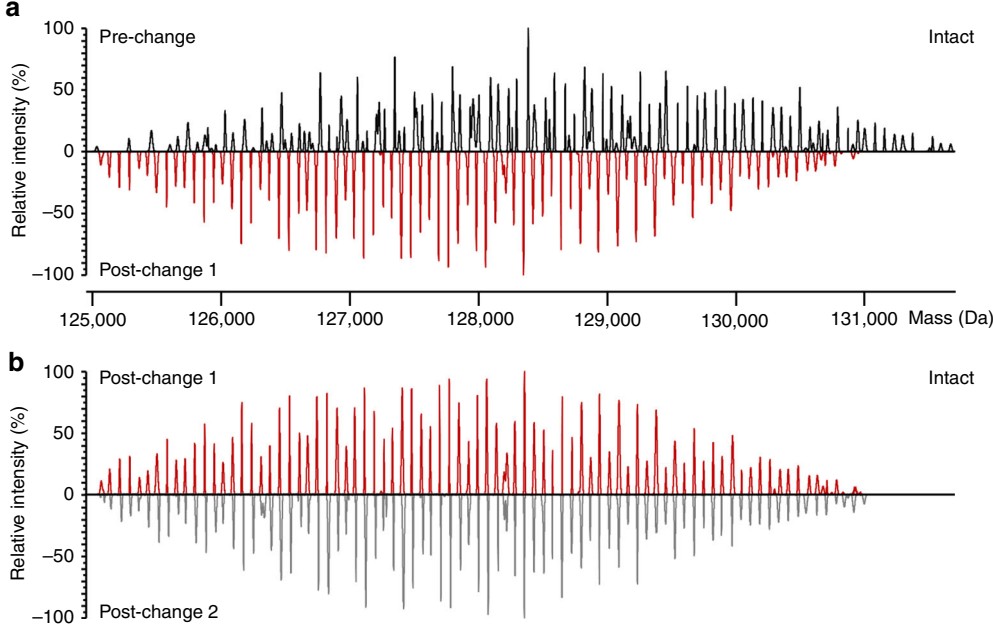

**Fig. 5** Comparison of pre- and post-change batches of Enbrel® upon native mass spectrometry at the intact protein level. **a** Mirror plot of deconvoluted mass spectra of Enbrel® EU pre- and post-change batch 1. **b** Mirror plot of deconvoluted mass spectra of Enbrel® EU post-change batches 1 and 2. Spectra were acquired at $R_{set} = 35,000$ at $m/z$ 200. Mass spectra were acquired with identical instrument settings (SID 100 eV; CE 25 eV)

**Assessment of N-glycosylation of the TNFR domain**. Although the spectrum of sialidase-treated Etanercept is less complex compared with the PNGase F digestion (Fig. 1d, e), attempts to assign N-glycoforms in an analogous strategy to that applied for O-glyco-forms at the whole protein level failed. This is mainly due the diversity of possible N-glycan structures and the large residual mass arising from six N-glycans present on the whole protein. Taking into account the O-glycoforms identified in the combined PNGase F/ sialidase digest (Fig. 2a), several possible compositions of the six desialylated N-glycans remained for each residual mass at this stage. Therefore, we pursued enzymatic dissection of Etanercept employing the IdeS protease to cleave the protein in the hinge region and subsequently separate the TNFR and Fc domains by affinity purification (Supplementary Fig. 4). This middle-up approach allowed for isolated analysis of the two domains, thereby considerably reducing the number of possible N-glycoform assignments.

In order to obtain information on N-glycan variants of TNFR, we first removed the O-glycans using O-glycosidase. As the employed enzyme is only active on unsubstituted O-glycan cores, the digest was performed in combination with sialidase (Supplementary Fig. 4). Therefore, the resulting peaks comprised only N-glycan variants lacking in sialic acid (Fig. 3a and Supplementary Fig. 6). The most abundant mass of 62,139 Da can be attributed to dimeric TNFR plus four N-glycan cores and additionally 10 hexose-, 10 HexNAc-, and two fucose-residues. As several structural isomers of this monosaccharide composition exist, unambiguous assignment of N-glycan structures is still not possible at this level.

To pinpoint the exact N-glycan structures present on the TNFR domain, we generated a site-specific library of N-glycans based on glycopeptides obtained after AspN digestion in a bottom–up glycopeptide analysis. Considering the site-specific fractional abundances of the identified glycopeptides (taking into account removal of sialic acids, see Supplementary Table 1), the most abundant glycoform (62,139 Da) most likely carries two A2G2 (68% fractional abundance at N149) and two A2G2F (83% fractional abundance at N171) N-glycans (for structures, see Supplementary Table 2). This N-glycan composition corresponds to four N-glycan cores and eight hexose-, eight HexNAc-, and

two fucose-residues, which leads us to conclude that the two remaining hexose- and HexNAc-residues, respectively, must be attributed to two O-glycan cores which were not removed by the O-glycosidase. In the same manner, most of the peaks present in the spectrum could unambiguously be ascribed to specific N-glycan combinations, i.e., N-glycoforms based on the fractional abundances obtained from glycopeptide data. In addition to the species comprising two residual O-glycan cores, variants bearing none or up to three O-glycan cores were detectable (Fig. 3a).

**Determination of N-glycosylation of the Fc domain**. Native MS of the affinity-purified Fc domain yielded molecular masses for the dimeric Fc domain (Fig. 3b and Supplementary Fig. 7), although no intermolecular disulfide bonds are present (Supplementary Fig. 4). Taking advantage of the corresponding tryptic glycopeptide abundances (N317, Supplementary Table 1) the most prominent gly-coforms were assigned to A2G0F/A2G0F (50,464.7 Da), A2G0F/ A2G1F (50,626.7 Da) and the isobaric variants A2G1F/A2G1F or A2G0F/A2G2F (50,788.7 Da) (Fig. 3b). Based on hit scores provided by MoFi, the abundance of the isobaric variants can be estimated at 70% for N-glycoform A2G1F/A2G1F and 30% for A2G0F/A2G2F. In addition, we observed C-terminal lysine variants at a relative abundance of ~24.3% (+Lys) and 2.8% (+2 Lys). This is in good agreement with the bottom–up data for the C-terminal tryptic peptides SLSLSPG (88.1%) and SLSLSPGK (11.9%), which corresponds to $0.881 \times 0.119 \times 2 = 21.0\%$ (+Lys) and $0.119^2 = 1.4\%$ (+2 Lys) variant in the Fc dimer.

**Integrating glycosylation data on the whole protein level**. Taking into account the N-glycan combinations identified on the subunit level (Fig. 3a, b), N-glycoforms lacking sialic acid can be assigned in sialidase/O-glycosidase treated Etanercept (Fig. 4a, Supplementary Fig. 4). Thus, the most abundant variant observed in Fig. 4a (112,732 Da) results from a combination of the most abundant TNFR N-glycoform (62,139 Da, Fig. 3a) and the second-most abundant Fc glycoform (50,627 Da, Fig. 3b) minus two water molecules incorporated upon proteolysis. In the same manner, other glycoforms of sialidase/O-glycosidase treated Etanercept can

be assigned upon integration of data on *N*-glycans for TNFR- and Fc domains, i.e., merging the assignments presented in Fig. 3a and 3b: six different *N*-glycan combinations substituted with up to three residual *O*-glycan cores could be identified (Fig. 4a). This annotation is consistent with the assignment of glycoforms based on a site-specific *N*-glycopeptide library (Supplementary Table 1), verifying coherence of data integration at different digestion levels.

Moving up one structural level, we attempted to assign *N*- and *O*-glycoforms after removal of sialic acid residues using the same approach. Upon integration of the most abundant *N*-glycan combinations annotated in the sialidase/*O*-glycosidase digest of Etanercept (Fig. 4a) and the *O*-glycoforms detected in the PNGase F/sialidase digest (Fig. 2a), it is possible to assign desialylated *N*- and *O*-glycans at the whole protein level (Fig. 4b). Annotation of Etanercept lacking sialic acid confirms that the two TNFR *N*-glycosylation sites are mainly occupied by 2 × A2G2 and 2 × A2G2F glycans, whereas the Fc *N*-glycosylation sites predominantly carry two different *N*-glycan structures, namely A2G0F and A2G1F. In addition, the prevalent *O*-glycoforms comprise 18, 19, and 20 *O*-glycan cores, respectively.

As far as glycosylation of intact (untreated) Etanercept is concerned, comparison of the spectra depicted in Fig. 1b and 1d clearly demonstrates the augmented complexity upon incorporation of sialic acid. Assignment of specific glycoforms on the whole protein level was generally ambiguous owing to insufficient resolution of multiple glycovariants, which differ only slightly in mass. Nevertheless, spectra of intact Etanercept displayed highly distinctive peak patterns, which prompted evaluation of the developed native MS method for assessment of glycan consistency in different Enbrel® production batches.

**Comparison of Enbrel® batches by native MS**. The analysis of intact Etanercept by native MS, as shown in Fig. 5, unravels between 92 and 122 distinct signals, providing a detailed and characteristic fingerprint of product heterogeneity. A change in the quality profile of Enbrel® production batches has previously been reported and was attributed to a decrease in A2G2F *N*-glycan structures as well as an increase in C-terminal lysine variants in post- compared with pre-change batches[28]. Intriguingly, we also observed differences in intact masses for pre- and post-change material (Fig. 5a), whereas spectra of two post-change batches were highly comparable (Fig. 5b). The observed differences could majorly be assigned to the *N*-glycan level, as native spectra of Enbrel® batches were more comparable after removal of *N*-glycans by PNGase F (Supplementary Fig. 9), consistent with previously published *N*-glycosylation differences[28].

**Discussion**

Mass spectrometric analysis of intact proteins under native-like conditions is a powerful tool to reveal coexisting glycoforms and may be supported by integration of middle–down and bottom–up proteomics data[22]. In contrast to released glycan or glycopeptide analysis, which yields information on an average glycoform, the context of glycosylation is conserved at the intact protein level, providing valuable information on true glycoform heterogeneity. Pursuing the systematic characterization of Etanercept, a highly glycosylated biopharmaceutical of 128 kDa mass, we have taken the concept of integrative mass spectrometric analysis to a new level of complexity. Our study reveals the coexistence of >100 glycoforms in a highly defined drug product. From a technical standpoint, the higher spatial resolution achieved upon mass spectrometric analysis of Etanercept in its native, folded state, is mandatory for the separation of the broad signal clusters arising from >100 proteoforms which differ in mass by as little as 20–40 Da (depending on molecular mass).

In excess of revealing unprecedented glycoform heterogeneity for a protein larger than 100 kDa, we aimed to assign specific glycovariants for the determined masses. Annotation of glycoforms to masses obtained for intact Etanercept using basic combinatorial approaches results in a vast number of ambiguous assignments: assuming a total of six *N*- and 24 *O*-glycans as well as 11 different *N*-glycan and three different *O*-glycan structures[24] for the most abundant proteoform of Etanercept (128,489 Da, Supplementary Fig. 3b), we obtain a total of 500 quadrillions ($11^6 \times 3^{24} \approx 5.00 \times 10^{17}$) potential glycan combinations to match the residual mass ($128,489 - 102,160 = 26,329$ Da). Additional constraints must be introduced in order to achieve unambiguous peak assignments. We therefore established an enzymatic dissection procedure beneficial in both reducing sample complexity as well as the residual mass to be explained.

Information on the number of *O*-glycan cores and their substitution by sialic acid could be deduced upon glycosidase digestion of intact Etanercept (Fig. 2). The extended search space arising from the structural diversity of *N*-glycans prompted us to integrate information on the glycan structures and abundances derived from glycopeptide data. Isolated analysis of affinity-purified Fc and TNFR revealed that core-fucosylated *N*-glycans lacking galactose predominate on the Fc domain, whereas on TNFR most *N*-glycans are galactosylated and structures lacking core fucose occur (Fig. 3). Using MoFi, systematic integration of data obtained for *O*-glycoforms, *N*-glycopeptides as well as *N*-glycoforms of TNFR and Fc domains, enabled annotation of masses up to the level of desialylated Etanercept. Supplementary Fig. 4 provides a roadmap for the transfer of information obtained from different digestion levels. Our strategy pursuing annotation of specific glycan structures on the whole protein level upon significant reduction of search space thus extends the previously described concept of assigning glycan compositions alone[22]. Finally, investigation at different structural levels enables localization of qualitative differences, exemplified by the higher comparability of Enbrel® batches upon removal of *N*-glycans as compared to the intact protein level, indicating that the observed differences mainly arise from *N*-glycosylation (Fig. 5, Supplementary Fig. 9).

Our study not only introduces a systematic strategy for glycoform assignment, but also assesses the current possibilities and limitations of native MS. Attempts to annotate glycan patterns at the level of intact Etanercept, including sialic acids, revealed that consolidation of glycoforms generated too many combinations for distinction by native MS, owing to overlapping isobaric variants or glycoforms of very small mass differences (for instance, two fucose- and one sialic acid-residue differ by only 1.0 Da; the mass difference between four *O*-glycan cores and five sialic acids is 5.0 Da). Thus, mass differences between intact and desialylated Etanercept cannot simply be explained by multiples of sialic acid masses, mainly because a signal observed in the spectrum of intact Etanercept does not generally represent a single, distinct glycoform. Intrinsic limitations of intact mass determination of large proteins include (i) the presence of unresolvable adduct-bound protein ions, (ii) overlapping signals of variants of small mass difference, which cannot be adequately resolved by MS (iii) the natural width of isotopic patterns of proteins containing >10,000 atoms[29]. As a consequence of the latter point, it is intrinsically impossible to resolve certain glycoforms as exemplified for an Etanercept PNGase F digest (Supplementary Fig. 10), even at isotopic resolution[29]. This limitation can be compensated, to some extent, through the integration of information obtained at different structural levels, e.g., taking into account relative abundances of the underlying *O*-glycoforms observed in the absence of sialic acid (Fig. 2a). The coherence in annotations clearly corroborates the feasibility of transferring glycoform information from the subunit level to the larger context of the whole protein.

In conclusion, the analysis of glycoforms by native MS combined with enzymatic dissection opens up new avenues in the characterization of heavily glycosylated biopharmaceuticals. Comprehensive information on glycoform heterogeneity, fast analysis with minimal sample preparation and product-characteristic fingerprints render our method highly attractive for the quality control of biologics as well as for comparability studies following changes in the manufacturing process. Assessment of biosimilarity and relative quantification of glycoforms at different levels of complexity will be further explored in the future.

## Methods

**Materials**. Cesium iodide (CsI) (Fisher Scientific, AC19282-010) was prepared at 2.0 mg mL$^{-1}$ in 30% acetonitrile (Fisher, Optima® LC-MS grade), 30% methanol (Fisher, Optima® LC-MS grade), and 40% water. Enbrel® (Amgen/Pfizer formerly distributed by Immunex/Wyeth) was purchased from the pharmacy, and stored and handled until analysis according to the manufacturer's instructions. Lot numbers and expiry dates are listed in Supplementary Table 3.

**Enzymatic digests and sample preparation**. Enbrel®, supplied in formulation buffer, was buffer exchanged into 100 mmol L$^{-1}$ aqueous ammonium acetate (p.a., Merck, Darmstadt, Germany) using Micro Bio-Spin P-6 columns (Bio-Rad, Vienna, Austria) with an exclusion cutoff of 6000 Da before MS analysis.

$N$-glycans were released from intact Etanercept using PNGase F from *Flavobacterium meningosepticum* (Roche Applied Science, Mannheim, Germany). For this purpose, 200 μg of the drug product were digested with 3.2 U of PNGase F (enzyme-to-substrate ratio 16 U mg$^{-1}$) in 15 mmol L$^{-1}$ Tris/HCl (Sigma Life Sciences, St. Louis, USA), pH 7.0, in a final volume of 100 μL. The sample was incubated for 17 h at 37 °C while shaking and then buffer exchanged into 100 mmol L$^{-1}$ aqueous ammonium acetate using a Micro Bio-Spin P-30 column (Bio-Rad) with an exclusion limit of 40,000 Da.

Intact Etanercept was desialylated using neuraminidase (sialidase) from *Arthrobacter ureafaciens* (Roche Applied Science). Two hundred micrograms of drug product were digested with 17 mU of sialidase (enzyme-to-substrate ratio 85 mU mg$^{-1}$) in 40 mmol L$^{-1}$ sodium acetate (puriss. p.a., Fluka Analytical, Steinheim, Germany) pH 5.0, in a final volume of 100 μL. The sample was incubated for 17 h at 37 °C while shaking and then buffer exchanged into 100 mmol L$^{-1}$ aqueous ammonium acetate using a Micro Bio-Spin P-30 column.

For release of $N$-glycans and sialic acids, 200 μg of drug product were digested with 3.2 U of PNGase F and 17 mU of sialidase in 15 mmol L$^{-1}$ Tris/HCl, pH 7.0, in a final volume of 100 μL. The sample was incubated for 17 h at 37 °C while shaking and then buffer exchanged into 100 mmol L$^{-1}$ aqueous ammonium acetate using a Micro Bio-Spin P-30 column.

IdeS digestion and affinity purification of the Fc domain were performed using the FragIT™ kit (Genovis, Lund, Sweden), containing FragIT™ and CaptureSelect™ spin columns, according to manufacturer's instructions. In brief, 400 μg of drug product was digested with immobilized FabRICATOR in a final volume of 100 μL of 20 mmol L$^{-1}$ ammonium bicarbonate (≥ 99.0%, Sigma Life Sciences). After 1.0 h incubation at 25 °C the sample was eluted and the column was flushed with 100 μL of 20 mmol L$^{-1}$ ammonium bicarbonate to achieve maximum recovery. The pooled eluates were loaded on a CaptureSelect™ column containing an Fc affinity matrix in order to separate the Fc and the TNFR domain. The flow-through of the purification step contained the TNFR domain whereas the Fc domain was eluted from the affinity matrix with 100 mmol L$^{-1}$ glycine (≥ 99%, Sigma Life Sciences), pH 3.0. The eluate was neutralized immediately with ammonium bicarbonate (1 mol L$^{-1}$). Twenty mmol L$^{-1}$ ammonium bicarbonate was used as cleavage and binding buffer.

For native MS of TNFR and Fc, 75 μL of the CaptureSelect™ flow-through or the eluate, respectively, were buffer exchanged with 100 mmol L$^{-1}$ aqueous ammonium acetate using a Micro Bio-Spin P-6 column. $N$-glycans were released from TNFR by addition of 750 U of PNGase F (New England BioLabs, Frankfurt am Main, Germany) to 13 μL of the CaptureSelect™ flow-through in a final volume of 25 μL of 20 mmol L$^{-1}$ ammonium bicarbonate. The sample was incubated for 17 h at 37 °C while shaking and then buffer exchanged into 100 mmol L$^{-1}$ aqueous ammonium acetate using a Micro Bio-Spin P-30 column. For release of sialic acids and $O$-glycans, TNFR was treated with a combination of sialidase from *A. ureafaciens* and $O$-glycosidase from *Enterococcus faecalis* (New England BioLabs). For this purpose, 1.0 μL of sialidase and 5.0 μL of $O$-glycosidase solution were added to 20 μL of CaptureSelect™ flow-through. After incubation for 17 h at 37 °C while shaking, the sample was buffer exchanged into 100 mmol L$^{-1}$ aqueous ammonium acetate using a Micro Bio-Spin P-30 column.

For glycopeptide analysis, Etanercept was reduced with 10 mmol L$^{-1}$ dithiothreitol (G-Biosciences, St. Louis, USA) in 6 mol L$^{-1}$ Gdn HCl (Sigma Life Sciences), 50 mmol L$^{-1}$ Tris/HCl (Sigma Life Sciences), 5.0 mmol L$^{-1}$ EDTA, pH 8 (Sigma Life Sciences) for 1.0 h at 37 °C, and alkylated with 20 mmol L$^{-1}$ iodoacetamide solution (G-Biosciences) for 1.0 h at 25 °C in the dark. Subsequently, the buffer was exchanged to 50 mmol L$^{-1}$ Tris/HCl, pH 8.0, using Amicon Ultra 0.5 mL centrifugal filters with an exclusion cutoff of 10 kDa

(Millipore, Darmstadt, Germany). For proteolytic digestion, 8.0 μg of Trypsin (Promega, Madison, USA) was added to 200 μg of Etanercept, and 1.0 μg of AspN (sequencing grade, Roche Applied Science) was added to 50 μg of Etanercept, respectively, followed by incubation for 17 h at 37 °C. Digestion was quenched by addition of TFA (LC-MS grade, Fluka Analytical).

**Native and denaturing mass spectrometry**. All protein solutions were analyzed at a concentration between 4 and 16 μmol L$^{-1}$ (assuming an average molecular mass of 128 kDa of intact Etanercept) in 100 mmol L$^{-1}$ aqueous ammonium acetate. Experiments were carried out on a Thermo Scientific™ Exactive™ Plus EMR mass spectrometer (Thermo Fisher Scientific, Bremen, Germany) equipped with a Thermo Scientific™ Nanospray Flex™ ion source. Samples were directly infused using conductive capillaries for static nanospray (catNo. ES387, Thermo Scientific™, San Jose, CA, USA) at a spray voltage of 1.2–1.6 kV. Ions were accumulated in the HCD cell in order to allow efficient trapping and desolvation. Trapping gas pressure and voltage offset on the transport multipoles and ion lenses were manually tuned. The scan range was set to $m/z$ 1000–10,000. S-lens RF level was set to 200. Instrument resolution, source induced dissociation, and collision energy were optimized for each sample in order to achieve best signal-to-noise ratios at highest possible resolution settings (typically between 17,500 and 70,000 at $m/z$ 200; higher resolution settings generally yielded noisier spectra of lower overall signal intensity). Comparison of different Enbrel® batches was performed at identical parameter settings. Spectra were acquired using rolling averaging in order to increase the signal-to-noise ratios. Depending on sample complexity, between 5000 and 15,000 microscans were averaged. The instrument was mass-calibrated based on signals detected from $m/z$ 392.7 to $m/z$ 11,304.7 using a solution of CsI.

**Mass spectrometric data processing and peak assignment**. Data were acquired using Xcalibur™ 3.0 software (Thermo Fisher Scientific) and raw spectra were deconvoluted to zero-charge using the ReSpect™ algorithm in the BioPharma Finder™ 1.0 SP1 software (Thermo Fisher Scientific). Deconvolution parameters were adjusted for each spectrum and the deconvoluted spectrum was verified by manual comparison with the raw data. Deconvoluted spectra were annotated using a computational tool, MoFi[26], which employs a two-stage search algorithm. In brief, the first search stage assigns monosaccharide/PTM compositions to each peak. Optionally, the software then compiles a hierarchical list of glycan combinations compatible with these compositions. The latter step requires a glycan library as derived from glycopeptide or released glycan analysis. A deviation of up to ±5 Da between experimental and theoretical average masses was accepted. MoFi also calculates a score for each assigned glycan combination by (1) multiplying the relative abundances of the included glycans and (2), for each peak, scaling the sum of scores of all alternative annotations to a value of 100. Hence, each score indicates the percentage of peak intensity that is explained by the respective glycoform. Scores for alternative $O$-core annotations in Fig. 2b were calculated likewise, using relative $O$-core abundances derived from the mass spectrum in Fig. 2a.

**Glycopeptide analysis by HPLC-MS/MS**. HPLC-MS/MS analyses of Etanercept digested with either trypsin or AspN were carried out on a Thermo Scientific™ Vanquish™ Horizon UHPLC system (Thermo Fisher Scientific, Germering, Germany) coupled with the Thermo Scientific™ Orbitrap Fusion™ Tribrid™ mass spectrometer (Thermo Fisher Scientific, San Jose, CA, USA) equipped with an HESI-II source. Five microliters of tryptic digest (1.0 mg mL$^{-1}$) and 10 μL of AspN digest (0.50 mg mL$^{-1}$) were injected, respectively. Chromatographic separation was achieved via a 150 × 2.1 mm Ascentis Express Peptide ES-C18 column (Supelco, Bellefonte, PE, USA) with 2.7 μm particle size operated at 55 °C. A gradient from 5.0 to 40% mobile phase B over 90 min with a flow rate of 200 μL min$^{-1}$ was applied with mobile phases A: water with 0.05% TFA (LC-MS grade, Fluka Analytical) and B: acetonitrile (gradient grade for LC, Merck) with 0.05% TFA.

Mass spectrometric data were acquired using a data dependent Top5 method comprising of a full scan followed by five MS/MS scans in each scan cycle. Full MS scans were acquired in the Orbitrap mass analyzer with the mass range set to $m/z$ 350–2000, a resolution setting of 60,000 at $m/z$ 200 with an automatic gain control (AGC) target value of 2e5 and a maximum injection time of 150 ms. HCD MS/MS scans were acquired in the Orbitrap mass analyzer with a resolution setting of 15,000 at $m/z$ 200, the precursor isolation window set to 3 Th, 30% normalized collision energy, an AGC target of 5e4, a maximum injection time of 250 ms and a fixed first mass of 120 Th. Data evaluation were performed with BioPharma Finder™ 1.0 SP1 software (Thermo Fisher Scientific). The following parameters were used: 200.0 S/N threshold, 5 ppm mass accuracy, 0.8 minimum confidence, specificity 'high' for the applied proteases, carbamidomethylation of the 29 cysteines as a fixed modification, and the built-in $N$-glycan repertoire for Chinese hamster ovary cell lines as variable modifications. Lys-variants were determined based on the tryptic peptides SLSLSPG and SLSLSPGK using a minimum confidence score setting of 0.0 in the BioPharma Finder™ software.

**Code availability**. MoFi software is freely available as source code from GitHub (https://github.com/cdl-biosimilars/mofi) and as a frozen executable for Linux and Windows (http://cdl-biosimilars.sbg.ac.at/).

**Data availability**. The following files used in this study are provided as ZIP archive (Supplementary Data 1) and on figshare (https://doi.org/10.6084/m9.figshare.6616106): raw spectra (in RAW and mzML format); MoFi parameter and output files; processed results and the corresponding processing scripts; glycopeptide data. See the readme file for further details. The data that support the findings of this study are available from the corresponding author upon request.

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

## Acknowledgements

The financial support by the Austrian Federal Ministry of Science, Research, and Economy and by a Start-up Grant of the State of Salzburg is gratefully acknowledged. We thank Rowan Moore from Thermo Fisher Scientific for critical proofreading of the manuscript, Silke Ruzek and Mona Schulte from Sandoz GmbH for technical support, Veronika Reisinger and Urs Lohrig from Global Drug Development, Novartis, Sandoz GmbH, and Stephane Houel from Thermo Fisher Scientific for scientific discussions. We acknowledge financial support by the Open Access Publication Fund of the University of Salzburg.

## Author contributions

T.W., I.C.F., K.S., J.H., S.S., and C.G.H conceived the study. T.W., K.S., I.C.F., and E.D. performed the experimental sections, data analysis, and interpretation. W.S. and S.S. established the software tool for glycan annotation. T.W., C.G.H., and K.S. drafted the manuscript. All authors reviewed the manuscript.

## Additional information

**Competing interests:** The authors declare the following competing financial interest(s): J. H. and I.C.F. are employees of Sandoz GmbH, K.S. and E.D. are employees of Thermo Fisher Scientific GmbH, which provide financial support for the Christian Doppler Laboratory for Innovative Tools for Biosimilar Characterization. The salary of T.W., W.S. and S.S. is fully funded, C.G.H.'s salary is partly funded by the Christian Doppler Laboratory for Biosimilar Characterization. Sandoz has obtained EMA and FDA approval for a biosimilar of Enbrel®. The authors declare no other competing financial interest.

