## [Peer Review File · Nature Communications]

Reviewers' comments:

Reviewer #1 (Remarks to the Author):

Wohlschlager et al. here report on the analysis of Etanercept using high-resolution native mass spectrometry combined with enzymatic treatment. As this is a highly relevant biopharmaceutical and exhibits a very complex glycosylation profile their work is a tour de force, brought about by the high-resolution mass spectra obtained with an Orbitrap EMR. The method is complemented also by bottom up mass spectrometry measurements. The authors highlight that the higher mass resolution achieved by native MS is a key parameter to facilitate the separation of different protein proteoforms differing from each other by only a small mass. Next, they applied enzymatic treatment of the sample in order to reduce sample complexity and molecular mass, which partially overcomes limitations resulting from the extremely high number of possible PTMs combinations and therefore difficulties with confident peak assignment. In general, this reviewer highly appreciates the presented work which demonstrates a unique ability of native MS measurements, which in combination with enzymatic processing of the sample can describe extreme glycoprotein micro heterogeneity. However, in its current form, the manuscript needs severe further improvements and detail especially due to incomplete data representation and relatively poor discussion.

Major critique:

Although the language in the manuscript is correct, I find it very difficult to read. The text is written very technically with many details, but the structure seems to be a bit chaotic. The Results part contains paragraphs and sentences, which should rather be in the discussion part. Major problem is the absence of clear tables (in supplementary material) containing vital information about detected and described glycoproteoforms of the analyzed sample and a list of detected glycopeptides. The table should contain a list of the all masses of “presented” glycoproteoforms with mass accuracy achieved, retention time (in case of glycopeptides), some score metrics etc. Result part could be probably structured better. For example, native spectra annotation could start from the enzymatically processed samples, which are “easier” to annotate. From that information, one could consecutively interpret more complicated spectra. Next, the discussion part needs to be improved. The manuscript is in many aspects alike the recent reported work of “Yang et al. Nat Commun 2016” and therefore this manuscript should be directly compared to this work stressing the novelty in this current work.

More detailed critique:

Results:

Page 5: "Limitations of intact mass determination...."

Page 9: "Due to the structural variability of N-glycans...."

Page 9: "Interestingly, masses obtained for the affinity-purified FC domain..."

Page 11: "Attempts to annotate glycan patterns revealed that..."

These are just some examples in the text, which should be rather in discussion part. In general, detail discussions on limitations of the approach and other difficulties should be removed from the results section, and mentioned in the discussion section.

Page 8:

"We determined the most abundant mass in the deconvoluted spectrum at 118,941. Taking into account...."

Mass of the most abundant species in the intact sample minus desialylated sample is 9548. (128,489 - 118,941 = 9548). $9548/291$ (mass of sialic acid) = 32.8 sialic acids. It seems they are missing somewhere 58 Da. Can they explain that? This also raises the question about mass accuracy and correctness of the spectra annotation in this manuscript in general. There is not accurate mass mentioned in the whole manuscript and I am forced to believe that everything is correct without chance to evaluate the data.

Fig. 2

The annotation of the native spectra is confusing and probably also not completely correct. Every peak assigned with different symbol corresponds to sialic acid variants of each glycoform. Peaks, which are in some certain ion series are certainly those glycoproteoforms with different number of sialic acids. However, these ion series are not in one "compact" cluster of ions consecutively differing by one sialic acid, but many of them are interrupted and continue from other masses. For example, a range of yellow squares (with annotated peak 18 O-cores + 22 Neu5Ac) ranging from 114000 Da to 116600 Da somehow continues also approx. 2kDa later. The same problem can be seen also for another peak series. I also would like to know, how the authors assumed that for example black half-moon series ends with 20 Neu5Ac. Next, the legend of the Fig. 2b is confusing. Every symbol corresponds only to a number of O-glycan cores, however it looks like it is the whole composition together with the number of sialic acids. Although it is mentioned in the Fig. description, the Fig. should be clearer.

I suggest to re-annotate the spectra based on the peak series and their abundances revealed in Fig. 2a. It shows that the dominant O-cores range from 16-21 and all other contribute only in minority. This would be actually also be consistent with the logic of the used approach.

Fig. 3

Here authors assigned with asterisks peaks as Na⁺ adducts. However, these assignments seem to be quite random and hardly justified.

Fig. 4

Is there no chance that HexNAcHex is not O-glycan but a polyLacNAc elongation on one of the

N-glycan branches?

Fig. 5

In the panel “a” authors observe differences between Pre-change and Post-change batches.

Authors should provide information about HCD energies since the main difference between the two batches seems to be a shift towards lower sialylation states. Higher HCD energies certainly result in loss of Neu5Ac on natively measured proteins.

Supplementary Fig. 9

There is unidentified peak series between 23+ and 22+ charge state 22+ and 21+, and 21+ and 20+. Can authors comment about it?

Supplementary table 2

A2Ga1G2F shows α -Gal structure, which is potentially immunogenic and undesirable on a biopharmaceutical product. This should be mentioned in the main text. Convincing MS/MS spectra corroborating structure must be included to support this specific structure.

Online Methods:

Why did authors use only HCD fragmentation technique? Would not EthcD as this is known to be better for glycopeptide analysis? Did the peptide data correlate well with the native MS measurements? Please discuss that in the text.

Discussion:

Reference 22. Should be discussed in the 1st paragraph as it is also focused on combining bottom up and native level data.

Second paragraph:

Limitations should be discussed more in detail including the issues with data analysis, mainly deconvolution:

The ReSpect algorithm does not produce a deconvoluted mass spectrum, it is rather deconvoluted mass value plotted as a function of ppm accuracy (“peak” width in deconvoluted spectra is determined by ppm accuracy rather than the real width of peak in original spectrum). This results in a weird observation wherein each peak in deconvoluted “spectrum” has different peak width. Based on my experience, “intensity value” obtained from deconvoluted spectra is highly irreproducible. This remains a major bottleneck in data analysis of complicated non-isotopically resolved spectra. The authors should discuss this

Reviewer #2 (Remarks to the Author):

The authors combine native mass spectrometry of an intact protein, native mass spectrometry of large fragments of the protein, and peptide analysis, to identify glycoforms on different sites in the protein, relative stoichiometries of different glycoforms, and glycoform composition of the

major forms of the intact protein. The protein they studied is heavily glycosylated (much more complicated than antibodies, which are the main proteins that have been studied by similar techniques), so presents a real challenge to characterize, but they have demonstrated what is achievable with a reasonable amount of work. While they did not achieve site-specific information about the O-glycosylation, they were able to deconvolute a fair amount of detail about the N-glycosylation stoichiometry on different sites.

They emphasize the advantage of native MS for spacial resolution of isotope clusters. Visually this is obviously significant, but even if isotope clusters overlap deconvolution software may be able to separate them; it would be good for the authors to comment on this, particularly as an Orbitrap will have higher resolution at the lower m/z observed in denatured analysis. Would this be easier / more effective if the data was acquired at higher resolution? Spectra in Figure 1 were acquired at resolutions ranging from 17.5K to 70K. If Fig 1b/1c were acquired at 70K then baseline separation of all of these peaks may have been obtained. I am not suggesting they need to re-acquire data at higher resolution, but mentioning why it was acquired at lower resolution would be appropriate.

Specific Comments:

I think the spectrum of the deconvoluted intact protein (Supplementary Fig. 3b) needs to be in the main part of the manuscript: this is the ultimate focus of the whole paper.

There should be a sentence that explains why versions with different numbers of lysines were expected; this is annotated without any explanation.

In the sialidase-treated data, how confident can the authors be that the removal was complete; i.e. they can assign two fucose over a residual NeuAc?

Responses to Reviewers:

Reviewer #1 (Remarks to the Author):

Wohlschlager et al. here report on the analysis of Etanercept using high-resolution native mass spectrometry combined with enzymatic treatment. As this is a highly relevant biopharmaceutical and exhibits a very complex glycosylation profile their work is a tour de force, brought about by the high-resolution mass spectra obtained with an Orbitrap EMR. The method is complemented also by bottom up mass spectrometry measurements. The authors highlight that the higher mass resolution achieved by native MS is a key parameter to facilitate the separation of different protein proteoforms differing from each other by only a small mass. Next, they applied enzymatic treatment of the sample in order to reduce sample complexity and molecular mass, which partially overcomes limitations resulting from the extremely high number of possible PTMs combinations and therefore difficulties with confident peak assignment. In general, this reviewer highly appreciates the presented work which demonstrates an unique ability of native MS measurements, which in combination with enzymatic processing of the sample can describe extreme glycoprotein micro heterogeneity. However, in its current form, the manuscript needs severe further improvements and detail especially due to incomplete data representation and relatively poor discussion.

Reviewer comment 1: *Major critique: although the language in the manuscript is correct, I find it very difficult to read. The text is written very technically with many details, but the structure seems to be a bit chaotic. The Results part contains paragraphs and sentences, which should rather be in the discussion part.*

Our reply 1: The manuscript has been fully revised with regard to structure and phrasing. Several sentences of the results section were moved to the discussion part as suggested by the reviewer. In order to provide an overview on the experimental workflow and the transfer of information for peak assignment, Supplementary Fig. 4 has been revised and now

illustrates the different structural levels described in this study. Phrasing was improved upon linguistic review by a native English speaker.

Reviewer comment 2: *Major problem is the absence of clear tables (in supplementary material) containing vital information about detected and described glycoproteoforms of the analyzed sample and a list of detected glycopeptides. The table should contain a list of the all masses of “presented” glycoproteoforms with mass accuracy achieved, retention time (in case of glycopeptides), some score metrics etc.*

Our reply 2: Files containing detected masses, relative abundances and assigned glycoforms have been added (Supplementary Data 1). The files also provide information on search parameters, mass deviation of assigned glycoforms with respect to the experimental mass, and hit scores where available. In case of glycopeptides, precursor m/z , product charge, mass error, total area, retention time and a confidence score provided by the software BioPharma Finder are listed.

Reviewer comment 3: *Result part could be probably structured better. For example, native spectra annotation could start from the enzymatically processed samples, which are “easier” to annotate. From that information, one could consecutively interpret more complicated spectra. Next, the discussion part needs to be improved.*

Our reply 3: In our intention, the order in spectra annotation does reflect an increase in sample complexity for O-glycans, N-glycans, and the combination of both. As the assignment of O- and N-glycans requires different enzymatic and computational approaches, an order purely based on spectral complexity is not coherent in our opinion. The logics we tried to elaborate in our argumentation was as follows: (1) the raw spectrum of intact Etanercept provides insight into the complexity of the molecule, illustrating the need for simplification at the experimental level (2) assignment of O-glycoforms and their sialylated variants is possible upon enzymatic deglycosylation at the whole protein level (3) assignment of N-glycans requires separate analysis of TNFR and Fc subunits and integration of glycopeptide data (4) information on both O- and N-glycans can be transferred to the whole protein level. We have outlined this strategy at the end of the section “Native mass spectrometry of Etanercept at the intact level” in order to clarify the sequence of experiments and data interpretation. Following the reviewer’s suggestion, we have revised and simplified the annotation in Figure 2b, which now better illustrates how glycan complexity increases with each level of enzymatic trimming. Finally, the discussion section has been completely re-written to emphasize the essential aspects of our workflow and to outline the limitations of the approach.

Reviewer comment 4: *The manuscript is in many aspects alike the recent reported work of “Yang et al. Nat Commun 2016” and therefore this manuscript should be directly compared to this work stressing the novelty in this current work.*

Our reply 4: This work of the group of A. Heck was mentioned (very briefly) in the first paragraph of the introduction in the initially submitted manuscript (ref. 22). We have now added information on the results presented by Yang et al. (first paragraph), our envisioned progress beyond that state-of-the-art (last paragraph of the introduction) and novel achievements described in this study (discussion).

Reviewer comment 5: *More detailed critique:*

Results:

Page 5: “Limitations of intact mass determination.....”

Page 9: “Due to the structural variability of N-glycans....”

Page 9: “Interestingly, masses obtained for the affinity-purified FC domain...”

Page 11: “Attempts to annotate glycan patterns revealed that...”

These are just some examples in the text, which should be rather in discussion part. In

general, detail discussions on limitations of the approach and other difficulties should be removed from the results section, and mentioned in the discussion section.

Our reply 5: We have added a paragraph to the first section of the results part to clarify the strategy behind the analysis at different structural levels. We have rearranged the text accordingly and shifted the sentences suggested by the reviewer to the discussion section. In addition, we have revised the titles of the subheadings to more clearly reflect the workflow employed for deciphering the multitude of glycoforms.

Reviewer comment 6: *Page 8: “We determined the most abundant mass in the deconvoluted spectrum at 118,941. Taking into account....” Mass of the most abundant species in the intact sample minus desialylated sample is 9548. (128,489 - 118,941 = 9548). 9548/291(mass of sialic acid) = 32.8 sialic acids. It seems they are missing somewhere 58 Da. Can they explain that? This also raises the question about mass accuracy and correctness of the spectra annotation in this manuscript in general. There is not accurate mass mentioned in the whole manuscript and I am forced to believe that everything is correct without chance to evaluate the data.*

Our reply 6: As outlined on page 7 as well as in Supplementary Figure 6 of our original manuscript, mass accuracy and mass resolution achieved for large biomolecules at high m/z are not sufficient to resolve the different glycoforms present under the peaks obtained for intact Etanercept. Therefore, we cannot simply derive the number of sialic acids from the mass difference observed in the spectra of the intact Etanercept and the desialylated molecule. Our viewpoint is corroborated by a recent publication on the limitations of mass resolution for distinguishing species of slightly differing mass, such as water- or sodium adducts (see Philip Lössl, Joost Snijder, Albert J. R. Heck, Boundaries of Mass Resolution in Native Mass Spectrometry, J. Am. Soc. Mass Spectrom., 2014, 25:906Y917), which is also discussed in our manuscript (reference 29 of the revised manuscript). Following the reviewer’s critique on the transparency of spectra annotation, files containing experimental and theoretical average masses have been added to the revised manuscript in order to document peak assignment and mass accuracy (Supplementary Data 1).

Reviewer comment 7: *Fig. 2. The annotation of the native spectra is confusing and probably also not completely correct. Every peak assigned with different symbol corresponds to sialic acid variants of each glycoform. Peaks, which are in some certain ion series are certainly those glycoproteoforms with different number of sialic acids. However, these ion series are not in one “compact” cluster of ions consecutively differing by one sialic acid, but many of them are interrupted and continue from other masses. For example, a range of yellow squares (with annotated peak 18 O-cores + 22 Neu5Ac) ranging from 114000 Da to 116600 Da somehow continues also approx. 2kDa later. The same problem can be seen also for another peak series.*

Our reply 7: The annotation of Fig. 2b has been revised with regard to clarity: taking into account relative abundances of the respective O-glycoforms observed in Fig. 2a (PNGase F+sialidase digest), only one, *i.e.* the predominant O-glycoform was assigned to each annotated peak. A file listing all theoretically possible glycoforms and their estimated abundances has been added (Supplementary Data 1, fig_2b/2b_7_figure_data.csv). The absence of certain peaks within sialic acid series is due to larger mass deviations than the accepted mass tolerance of +/- 5 Da. This may be due to overlapping proteoforms, which were not resolved. In the new version of Fig. 2b this is only the case for one glycoform, which is now also discussed in the text.

Reviewer comment 8: *I also would like to know, how the authors assumed that for example black half-moon series ends with 20 Neu5Ac.*

Our reply 8: As discussed in the previous paragraph, the absence of certain peaks within sialic acid series is due to larger mass deviations than the accepted mass tolerance of +/- 5 Da. This may be due to overlapping proteoforms, which were not resolved. Considering the low relative abundance of the black half-moon O-glycoform corresponding to 22 O-glycan cores in Fig. 2a, this symbol has been completely removed from the annotation in Fig. 2b. Based on the relative abundance of each glycoform, a hit score was introduced in order to transfer this information between different digestion levels. For clarity, the revised version of Fig. 2b only shows highly abundant O-glycoforms, *i.e.* the most probable variants. A file listing all theoretically possible glycoforms and their estimated abundances has been added (Supplementary Data 1, fig_2b/2b_7_figure_data.csv).

Reviewer comment 9: *Next, the legend of the Fig. 2b is confusing. Every symbol corresponds only to a number of O-glycan cores, however it looks like it is the whole composition together with the number of sialic acids. Although it is mentioned in the Fig. description, the Fig. should be clearer.*

Our reply 9: Figure 2b has been revised as described in replies 7 and 8. Additionally, the number of incorporated sialic acids is now indicated above each annotated peak.

Reviewer comment 10: *I suggest to re-annotate the spectra based on the peak series and their abundances revealed in Fig. 2a. It shows that the dominant O-cores range from 16-21 and all other contribute only in minority. This would be actually also be consistent with the logic of the used approach.*

Our reply 10: This is a very helpful suggestion and Figure 2b was changed accordingly. As mentioned in reply 8, a hit score based on the relative abundance of each glycoform was introduced in order to transfer this information between different digestion levels. A peak list containing all possible glycoform assignments with their hit scores has been added as supplementary data.

Reviewer comment 11: *Fig. 3. Here authors assigned with asterisks peaks as Na⁺ adducts. However, these assignments seem to be quite random and hardly justified.*

Our reply 11: The mass of 22.99 Da corresponding to one Na⁺ ion was included in the list of possible modifications for the annotation of glycoforms. The Na⁺ adducts indicated are within the accepted mass accuracy and occur in tandem with an M+H⁺ peak at a relative abundance of around 10%-13%. This pattern of a main M+H⁺ peak and a Na⁺ peak of lower intensity is characteristic in case of adduct formation and supports the annotation of the respective peaks.

Reviewer comment 12: *Fig. 4. Is there no chance that HexNAcHex is not O-glycan but a polyLacNAc elongation on one of the N-glycan branches?*

Our reply 12: PolyLacNAc was neither detected in glycopeptide analysis (this study) nor in the analysis of released N-glycans (Houel S. et al., Anal. Chem., 2014), the latter representing the gold standard in glycan analysis. This structure was therefore not included in the applied N-glycan library (compare Supplementary Table 1).

Reviewer comment 13: *Fig. 5. In the panel "a" authors observe differences between Pre-change and Post-change batches. Authors should provide information about HCD energies since the main difference between the two batches seems to be a shift towards lower sialylation states. Higher HCD energies certainly result in loss of Neu5Ac on natively measured proteins.*

Our reply 13: As outlined in the methods section of the manuscript, comparison of different Enbrel[®] batches was performed at identical instrument settings. Mass spectra shown in Fig.

5 were all acquired with fragmentation energy settings of SID 100 eV and CE 25 eV, ruling out the possibility that the observed difference in intact masses is due to an experimental artifact. Information on fragmentation energy settings has been added to the corresponding figure legend. In addition, raw files containing information on instrumental settings will be made publicly available. With regard to fragmentation energies, settings were initially optimized by systematically increasing fragmentation energies while following the formation of characteristic glycan fragment ions at low m/z . Fragmentation energies were chosen in order to achieve a compromise between maximum desolvation and minimum glycan fragmentation.

The observed shift towards lower masses for the two post-change batches compared to the pre-change batch is in-line with a previous report describing a decrease in variants containing the N-glycan G2F and an increase in G0F glycoforms, corresponding to lighter variants due to a lower number of incorporated galactoses (Schiestl M. et al., Nat. Biotechnol., 2011).

Reviewer comment 14: *Supplementary Fig. 9. There is unidentified peak series between 23+ and 22+ charge state 22+ and 21+, and 21+ and 20+. Can authors comment about it?*

Our reply 14: Charge states of this species were labelled in blue in Supplementary Fig. 8b of the revised manuscript (Supplementary Fig. 9 of the submitted manuscript) and can be assigned to a truncated Etanercept dimer lacking amino acids 1 to 186 of one TNFR chain within a mass accuracy of ± 30 ppm.

Reviewer comment 15: *Supplementary table 2. A2Ga1G2F shows α -Gal structure, which is potentially immunogenic and undesirable on a biopharmaceutical product. This should be mentioned in the main text. Convincing MS/MS spectra corroborating structure must be included to support this specific structure.*

Our reply 15: The reviewer is correct that identification of the immunogenic A2Ga1G2F structure requires further consideration. As the identification was solely based on intact glycopeptide mass (upon identification by MS/MS in a different Enbrel[®] batch), the respective structure has been eliminated from Supplementary Tables 1 and 2. The revised tables now contain only structures for which reliable MS/MS spectra were acquired. Because of the low abundance of the eliminated glycan structures ($\leq 1.7\%$), the resulting effect on glycoform annotation at the subunit and whole protein level is minimal.

Reviewer comment 16: *Online Methods: Why did authors use only HCD fragmentation technique? Would not EthcD as this is known to be better for glycopeptide analysis? Did the peptide data correlate well with the native MS measurements? Please discuss that in the text.*

Our reply 16: The reviewer is correct that for in-depth analysis aiming at maximum sequence coverage and identification of minor glycoforms, EThcD is the method of choice. In this study, the purpose of glycopeptide analysis was the generation of a database containing the main N-glycan structures. Therefore, HCD was applied as a generic method fit for purpose. The identified N-glycopeptides are well in accordance with previously published structures identified by released N-glycan analysis, which is the gold standard for glycan analysis (Houel S. et al., Anal. Chem., 2014). A comparison of N-glycan structures and their relative abundances as identified by glycopeptide and released glycan analysis is shown below (released glycans up to a fractional abundance of 1.0% were derived from Houel, S. et al. 2014; peptide mapping data originate from the current study).

Glycans including sialic acids				Desialylated glycans			
Released glycans		Peptide mapping		Released glycans		Peptide mapping	
Glycan	%	Glycan	%	Glycan	%	Glycan	%
A2G0F	21.9	A2S1G1F	21.1	A2G2F	35.3	A2G2F	39.7
A2S1G1F	20.4	A2G0F	17.1	A2G0F	21.9	A2G2	22.6
A2G1F	14.2	A2G1F	15.1	A2G2	19.1	A2G1F	17.2
A2S1G1	12.6	A2S1G1	14.0	A2G1F	17.4	A2G0F	17.1
A2G2F	7.8	A2G2F	11.7	M5	3.3	M5	2.6
A2S2F	7.0	A2S2F	6.9	A2G0	1.0	A2G0	0.8
A2G2	3.9	A2G2	5.5				
M5	3.3	A2S2	3.2				
A2S1G0F	3.2	M5	2.6				
A2S2	2.6	A2S1G0F	2.2				
A2G0	1.0	A2G0	0.8				

The main advantage of glycopeptide data is the information on glycosylation site occupancy, which is considered in the annotation of subunit and whole protein spectra. In order to assess correlation of glycopeptide data with native MS measurements, we compared a theoretical spectrum simulated based on our desialylated glycan library (Supplementary Table 1) with the experimental spectrum obtained for Etanercept lacking sialic acid and O-glycans (Fig. 4a). The mirror plot shown below demonstrates coherence of the theoretical spectrum based on glycopeptide data and the experimentally derived spectrum: almost all experimentally detected masses correspond to a theoretically calculated mass, providing evidence that the applied N-glycan library is adequate for annotation of native mass spectra. Nevertheless, it should be noted that sialylated N-glycans were not evaluated in the same way, as annotation of native mass spectra of sialylated Etanercept was not possible due to the limits in mass resolution discussed in the manuscript.

Reviewer comment 17: Discussion: Reference 22. Should be discussed in the 1st paragraph as it is also focused on combining bottom up and native level data.

Our reply 17: As outlined in reply 4, we have added more detailed information on the results presented in reference 22 as well as a comparison with the findings described in this study (introduction and discussion sections).

Reviewer comment 18: *Second paragraph: Limitations should be discussed more in detail including the issues with data analysis, mainly deconvolution: The ReSpect algorithm does not produce a deconvoluted mass spectrum, it is rather deconvoluted mass value plotted as a function of ppm accuracy (“peak” width in deconvoluted spectra is determined by ppm accuracy rather than the real width of peak in original spectrum). This results in a weird observation wherein each peak in deconvoluted “spectrum” has different peak width. Based on my experience, “intensity value” obtained from deconvoluted spectra is highly irreproducible. This remains a major bottleneck in data analysis of complicated non-isotopically resolved spectra. The authors should discuss this*

Our reply 18: We fully share the concerns of this reviewer regarding deconvolution of isotopically unresolved ESI spectra, especially concerning the relative peak intensities of the deconvoluted species. Nevertheless, we have clear evidence that the ReSpect algorithm yields a very good estimate for the uncharged centroid mass from the raw spectra, which is the basis of our peak annotation. The requested information regarding the use of the ReSpect algorithm has been added to the Figure legend of Supplementary Fig. 3 also mentioning that the peak widths shown in deconvoluted spectra represent the error of mass calculation from multiply charged species.

Supplementary Fig. 3 also shows a back-to-back comparison of raw and deconvoluted data, demonstrating the correlation of relative intensities in the deconvoluted spectrum with the intensities observed in the raw spectrum. In addition, a list containing the obtained deconvoluted masses and their underlying m/z values with the respective signal intensities has been added as supplementary data. We agree that intensity values to some degree depend on parameter settings in the deconvolution software. As outlined in the methods section of the manuscript, deconvolution parameters were adjusted and deconvoluted spectra were verified by manual comparison with the raw data.

Reviewer #2 (Remarks to the Author):

The authors combine native mass spectrometry of an intact protein, native mass spectrometry of large fragments of the protein, and peptide analysis, to identify glycoforms on different sites in the protein, relative stoichiometries of different glycoforms, and glycoform composition of the major forms of the intact protein. The protein they studied is heavily glycosylated (much more complicated than antibodies, which are the main proteins that have been studied by similar techniques), so presents a real challenge to characterize, but they have demonstrated what is achievable with a reasonable amount of work. While they did not achieve site-specific information about the O-glycosylation, they were able to deconvolute a fair amount of detail about the N-glycosylation stoichiometry on different sites.

Reviewer comment 19: *They emphasize the advantage of native MS for spatial resolution of isotope clusters. Visually this is obviously significant, but even if isotope clusters overlap deconvolution software may be able to separate them; it would be good for the authors to comment on this, particularly as an Orbitrap will have higher resolution at the lower m/z observed in denatured analysis. Would this be easier / more effective if the data was acquired at higher resolution? Spectra in Figure 1 were acquired at resolutions ranging from 17.5K to 70K. If Fig 1b/1c were acquired at 70K then baseline separation of all of these peaks may have been obtained. I am not suggesting they need to re-acquire data at higher resolution, but mentioning why it was acquired at lower resolution would be appropriate.*

Our reply 19: The resolving power of the applied Exactive™ Plus EMR instrument is not sufficient to resolve isotope clusters of molecules of the mass and complexity employed in our study. We would need a resolving power of more than half a million to be able to resolve the isotopic pattern of intact Etanercept, which is not available on the Exactive™ Plus EMR instrument. In general, the detection process of protein ions in the Orbitrap mass analyzer is limited by the signal decay. The optimal resolution setting generally depends on several

physical parameters: the molecular mass, charge state and collisional cross section of the ions as well as the pressure regime inside the mass spectrometer, relating to the pressure during trapping and ion detection. These factors impact the stability, i.e. the lifetime of analyte ions in the Orbitrap, which is determined by decay upon dephasing due to scattering and reduction of radial and axial amplitudes of oscillations due to non-fragmenting collisions. As large and heavily modified protein ions cannot be isotopically resolved, it is generally advised to start with the lowest available resolution setting that translates directly into a short transient detection time. Increasing the resolution setting by one or two steps may sometimes help to resolve adducts. However, increasing the resolution setting even further will result in decreased spectral quality when the signal transient is acquired beyond the point of complete signal decay thus resulting in the accumulation of noise. We generally recorded spectra at the highest possible resolution settings aiming for optimal spectral quality. This information has been added to the methods section. With regard to the raw spectra shown in Fig. 1, higher resolution settings could be applied to samples of lower complexity because of higher initial signal intensities and slower signal decay as compared to samples of increased complexity.

Reviewer comment 20: *Specific Comments: I think the spectrum of the deconvoluted intact protein (Supplementary Fig. 3b) needs to be in the main part of the manuscript: this is the ultimate focus of the whole paper.*

Our reply 20: According to Reviewer comment 18, we have added detailed information on the ReSpect algorithm applied for deconvolution. As we show deconvoluted spectra of intact Etanercept in Fig. 5, we suggest to keep the zero charge spectrum of Fig. 1b including additional information on the deconvolution in the supplementary material.

Reviewer comment 21: *There should be a sentence that explains why versions with different numbers of lysines were expected; this is annotated without any explanation.*

Our reply 21: This information including a reference has been added to the section "Determination of Etanercept O-glycoforms upon removal of N-glycans".

Reviewer comment 22: *In the sialidase-treated data, how confident can the authors be that the removal was complete; i.e. they can assign two fucose over a residual NeuAc?*

Our reply 22: Upon treatment of Etanercept with both sialidase and PNGase F, no delta masses corresponding to sialic acid residues were observed (Figure 2a), indicating that sialidase digestion was complete. In case of the PNGase F digest (Figure 2b), delta masses can unambiguously be assigned to Neu5Ac, as no fucose-residues are present upon removal of N-glycans.

Reviewer 1

Although the authors did substantially improve the manuscript it still does demonstrate to little improvement from earlier work.

More critically novelty is further diminished by a recent paper

Protein J. 2018 Feb 6. doi: 10.1007/s10930-018-9757-y. PMID: 29411222

that analyzed Enbrel and its biosimilar Altebrel. In this study they demonstrated intact mass measurement, glycopeptide and released glycan analysis and Middle up measurement of IdeS treated samples. The major difference is that the intact mass was performed with a Q-TOF.

Compared to that work the current paper does not add much

Reviewer 2

The revised manuscript has addressed my comments.

Reviewer 1

Although the authors did substantially improve the manuscript it still does demonstrate to little improvement from earlier work.

More critically novelty is further diminished by a recent paper

Protein J. 2018 Feb 6. doi: 10.1007/s10930-018-9757-y. PMID: 29411222

that analyzed Enbrel and its biosimilar Altebrel. In this study they demonstrated intact mass measurement, glycopeptide and released glycan analysis and Middle up measurement of IdeS treated samples. The major difference is that the intact mass was performed with a Q-TOF.

Compared to that work the current paper does not add much

Our reply: The study referred to by the reviewer was published while our manuscript was under review at Nature Communications. We are perplexed by the reviewer's comment as the overlap of our study with the mentioned publication is minimal: Montacir O. *et al.* present data on glycopeptides and released glycans of Etanercept, an approach that was previously published by Houel S. *et al.* (2014) and was referenced in our manuscript (ref. 24). Low resolution mass spectra of desialylated Etanercept as well as TNFR and Fc/2 subunits showed only very few molecular species. At the best, they provide an estimation of the molecular mass but neither reveal glycoform heterogeneity nor facilitate assignment of complex glycoforms, which is the main point of our manuscript (compare Fig. 1 and 2 in Montacir O. *et al.* to Fig. 4b and 3 in our manuscript, respectively). Mass spectra of the intact molecule were not provided. We therefore do not see any conflict with our study.